# Is Early Surgical Intervention Necessary for Acute Neonatal Humeral Epiphyseal Osteomyelitis: A Retrospective Study of 31 Patients

**DOI:** 10.3390/children9040527

**Published:** 2022-04-07

**Authors:** Yun Gao, Ruikang Liu, Saroj Rai, Qingtuan Liang, Yuan Liu, Xiaoliang Xiao, Pan Hong

**Affiliations:** 1Department of Orthopaedic Surgery, Zhuhai Center for Maternal and Child Health Care, Zhuhai 519000, China; gaoyun7268@163.com (Y.G.); liangqt2006@163.com (Q.L.); liuyuan919@163.com (Y.L.); xxl831210@163.com (X.X.); 2Department of Endocrinology, Union Hospital, Tongji Medical College, Huazhong University of Science and Technology, Wuhan 430022, China; rickylrk@163.com; 3Department of Orthopaedics and Trauma Surgery, Blue Cross Hospital, Tripureswor, Kathmandu 44600, Nepal; mesaroz@outlook.com; 4Department of Orthopaedic Surgery, Union Hospital, Tongji Medical College, Huazhong University of Science and Technology, Wuhan 430022, China

**Keywords:** neonatal humeral epiphyseal osteomyelitis, surgery, septic arthritis

## Abstract

Objective: To review the treatment experience of neonatal humeral epiphyseal osteomyelitis retrospectively. Study design: Retrospective cohort study of infants with neonatal humeral epiphyseal osteomyelitis. Patients were divided into conservative group and surgical group, and the surgical group was subdivided into early and delayed surgical group. Results: In total, there were 7 patients in the conservative group and 24 in the surgical group. The length of hospital stay and intravenous course of antibiotic therapy were both significantly shorter in the surgical group (*p* < 0.001). The full recovery rate was also higher in the surgical group (83.3%) than the conservative group (14.3%) (*p* < 0.001). Early surgery group (*n* = 14) had an insignificantly higher positive rate of pus/aspirate culture and full recovery rate than delayed surgery group (*n* = 10). Conclusion: Surgical treatment for neonatal humeral epiphyseal osteomyelitis demonstrated significantly higher rates of positive culture for the pathogen, a shorter course of intravenous oral antibiotics, and lower incidence of growth abnormality than conservative treatment. In our institution, most of culture outcome Gram-positive bacteria, and early surgical treatment was recommended with better outcome than delayed surgical group. Empirical antibiotics should be tailored to the epidemiological characteristics of local virulent bacteria.

## 1. Introduction

Neonatal osteomyelitis is a rare entity, and the incidence is around 0.1–0.3% of hospital admissions in developed countries [1,2]. If not treated promptly, it might result in catastrophic consequences, including sepsis, chronic infection, growth disturbance, and even multiple organ failure [3,4]. Previous reports suggested that the appropriate antibiotic treatment alone suffices for 90% of patients with acute hematogenous osteomyelitis [5,6]. Therefore, in clinical practice, the primary choice for neonatal osteomyelitis was empiric antibiotic treatment followed by regimen readjustment according to the bacterial culture or PCR results. Recently, certain studies recommended surgical treatment, mostly for patients of concomitant septic arthritis [7,8,9,10]. However, there was no consensus on the treatment strategy for children with epiphyseal osteomyelitis. This study aims to review the clinical outcomes of surgical intervention vs. conservative treatment for neonatal humeral epiphyseal osteomyelitis.

## 2. Materials and Methods

All 31 neonates diagnosed with epiphyseal osteomyelitis between January 2010 and January 2018 at Union Hospital, Tongji Medical College, Huazhong University of Science and Technology, and Zhuhai Center for Maternal and Child Health Care, were retrospectively reviewed.

Neonate (<28 days old) diagnosed with the humeral epiphyseal osteomyelitis treated either surgically or conservatively and followed-up for at least 12 months with complete medical records were included. All included patients focused on proximal humeral epiphyseal osteomyelitis. They were treated according to the diagnostic algorithm (see Figure 1). The neonate with multifocal infections in the musculoskeletal system was excluded.

Since this study was a retrospective investigation, the Ethics Committee of Tongji Medical College, Huazhong University of Science and Technology ruled that no formal ethics approval was required.

### 2.1. Tests and Examinations

Radiological examinations, including X-ray of chest and shoulder and magnetic resonance imaging (MRI) of the involved shoulder, were routinely performed as an emergency to improve the detection of concomitant septic arthritis and osteomyelitis. Blood tests, including routine blood test, C-reactive protein (CRP), erythrocyte sedimentation rate (ESR), were also performed. Other investigations included the blood culture, pus culture, and culture from aspirate and necrotic tissues.

#### 2.1.1. Conservative Method

Indications of conservative treatment were the patient presenting to the hospital after more than 1 week of onset of symptoms or surgical intervention rejected by the parents following improvements (clinical and laboratory inflammatory markers) after antibiotic therapy (see Figure 2).

#### 2.1.2. Surgical Intervention

Surgery was indicated if the patient did not show any improvement in clinical symptoms (such as significant temperature fluctuation) and laboratory markers (such as C-reactive protein) after empiric antibiotic therapy. Besides, early surgery was recommended in cases where MRI demonstrated concomitant septic arthritis. It was divided into two groups, i.e., early surgery group and delayed surgery group. Early surgery was regarded as surgery performed within 72 h of the onset of symptoms. Delayed surgery group was regarded as surgery performed after 72 h. Most of our patients were transferred from the local hospital. They had all already received variable doses of antibiotics (see Figure 3).

#### 2.1.3. Antibiotic Therapy

The empirical intravenous (IV) antibiotic regimen included Oxacillin (25 mg/kg, Q6h) alone or Cefuroxime (50–75 mg/kg/d, Q8h) combined with Clindamycin (20–30 mg/kg/d, Q8h). Sometimes, Linezolid (10 mg/kg, Q8h) was added to target Gram-positive bacteria. An appropriate antibiotic is adjusted after the culture and sensitivity report. Empirical antibiotic therapy is continued if the culture and sensitivity report was negative. After the resolution of clinical symptoms and normalization of inflammatory markers, oral antibiotics were continued for 2–4 weeks.

#### 2.1.4. Surgical Technique

The patient was placed supine on the operative table under general anesthesia. An anterior approach to the shoulder was adopted to expose the capsule and the proximal humerus. The pus and necrotic tissues were removed and sent for culture and sensitivity test. The incision was secured with vacuum-assisted closure (VAC) dressing. The secondary closure of the wound was performed within 1–2 weeks.

#### 2.1.5. Follow-Up at Out-Patient Visit

Follow-up visits were scheduled every 2 weeks for the first 3 months, then 6 months, 12 months, and yearly after that. Routine blood tests, CRP, and ESR were evaluated at each of the follow-up visits. X-ray was also taken in each of the follow-up visits. Clinical examinations, including gross appearance, range of motion (ROM) of the shoulder joint, and length of upper extremities, were also assessed and recorded at every visit.

Full recovery required the following criteria to be met: (1) unrestricted ROM of shoulder joint without pain; (2) no growth disturbance; (3) appearance of normal X-ray of the infected shoulder joint.

### 2.2. Data Collection

Baseline information for patients, including age, sex, duration from onset to admission, duration from onset to surgery, length of antibiotic treatment, length of hospital stay, and other predisposing factors, were recorded from the hospital database. Other laboratory parameters, including blood investigations and inflammatory markers, blood and pus/aspirate cultures, were also recorded from the hospital database. Both the clinical and radiological data were recorded in every follow-up visit.

### 2.3. Statistical Analysis

Statistical analysis was performed using SPSS (SPSS Inc, Chicago, IL, USA). All descriptive data were presented as the mean ± SD. A *p*-value of <0.05 was regarded as statistically significant.

## 3. Results

In total, 31 patients were included in this study (study flowchart in Figure 4). There were 7 patients (4 male, 3 female) in the conservative group and 24 (16 male, 8 female) in the surgical group (see Table 1). All patients were followed-up for more than 12 months, ranging from 14 to 68 months. The duration from onset to admission was significantly longer in the conservative group (11.0 ± 1.9 days) than the surgical group (2.4 ± 1.6 days) (*p* < 0.001). There were no significant differences between the conservative and surgical groups concerning age, sex, routine blood tests (leukocyte, neutrophil, lymphocyte and platelet counts), CRP, and ESR level.

As shown in Table 2, no statistically significant differences were observed in the two groups regarding the blood culture, as two (28.6%) cases in the conservative group and five (20.8%) cases in the surgical group had positive blood culture (*p* = 0.68). However, 17 (70.8%) of the patients in the surgical group had positive pus/aspirate culture, whereas only one (14.3%) case in the conservative group had positive pus/aspirate culture (*p* < 0.001). The length of hospital stay was significantly shorter in the surgical group (24.8 ± 2.8, d) than in the conservative group (45.6 ± 2.4, d) (*p* < 0.001). The IV course of antibiotic therapy was significantly shorter in the surgical group (24.8 ± 2.8, d) than in the conservative group (45.6 ± 2.4, d) (*p* < 0.001). The oral course of antibiotic therapy was significantly shorter in the surgical group (19.1 ± 5.3, d) than in the conservative group (27.7 ± 1.2, d) (*p* < 0.001). The full recovery rate was higher in the surgical group (20/24, 83.3%) than in the conservative group (1/7, 14.3%) (*p* < 0.001). There was a significant difference between the conservative and surgical group concerning limb length discrepancy (0.7 ± 0.5 vs. 0.4 ± 0.2, CM), ROM of the shoulder joint, and appearance of normal X-ray (14.3% vs. 83.3%) (*p* < 0.001). Only one patient in the conservative group was completely recovered with aggressive and appropriate antibiotic therapy. In this patient, the diagnosis was made based on a culture and sensitivity test. MRI was not routine in our follow-up to ascertain the percentage of physical injuries in patients without full recovery.

The surgical group was divided into two subgroups (see Table 3): 14 patients (male 9, female 5) in early surgery and 10 patients (male 7, female 3) in delayed surgery. The duration from onset to surgery was shorter in early surgery (1.7 ± 1.2, d) than delayed surgery (4.4 ± 0.8, d) (*p* < 0.001). There were no significant differences between early and delayed surgical groups concerning age, sex, routine blood tests (leukocyte, neutrophil, lymphocyte, and platelet counts), CRP, ESR level, and other predisposing factors.

As shown in Table 4, 11 (78.6%) patients in the early surgery group had a positive pus/aspirate culture, whereas only six (60.0%) patients in the delayed surgery group had positive pus/aspirate culture (*p* = 0.316). *Staphylococcus aureus* was the most commonly detected microorganism. The length of hospital stay was significantly shorter in the early surgery group (23.4 ± 2.4, d) than in the delayed surgery group (26.9 ± 1.9, d) (*p* < 0.001). The IV course of antibiotic therapy was significantly shorter in the early surgery group (23.4 ± 2.4, d) than in the delayed surgery group (26.9 ± 1.9, d) (*p* < 0.001). The oral course of antibiotic therapy was significantly shorter in the early surgery group (15.6 ± 3.5, d) than in the delayed surgery group (24.1 ± 2.7, d) (*p* < 0.001). The full recovery rate was higher in early surgery (12/14, 85.7%) than in the delayed surgery group (8/10, 80%) (*p* = 0.739). There was no significant difference between early surgery and delayed surgery groups concerning limb-length discrepancy (0.4 ± 0.17 vs. 0.35 ± 0.21, CM), ROM of shoulder joint, and appearance of normal X-ray (85.7% vs. 80.0%).

## 4. Discussion

Surgical intervention led to a higher rate of full recovery, shorter course of intravenous antibiotics and better shoulder ROM. In addition, the percentage of positive pus/aspirate culture was higher in the surgical group (70.8%). Early surgical intervention was recommended, with a shorter course of intravenous antibiotics and lower incidence of growth abnormality than delayed surgical intervention.

The infection of bone and joint is relatively rare in neonates [11]. Predisposing factors include perinatal asphyxia, chorioamnionitis, and prolonged rupture of membranes [12]. However, only 15% of the patients in our study had clear predisposing factors. Early diagnosis of neonatal infection is mandatory for better outcomes; therefore, both the surgeon and pediatrician should always suspect the possibility of osteomyelitis.

CRP and ESR are routine inflammatory markers in the evaluation of osteomyelitis [4,13]. However, these inflammatory markers are nonspecific. CRP changes more rapidly than ESR and is usually the preferred test to monitor the course of illness [4]. In our study, all patients displayed elevated CRP levels at admission, but not all patients manifested an increased number of leukocyte count, consistent with previous reports [3,4].

Identifying the causative pathogen is imperative for the proper treatment of infection [4,14]. However, the blood culture for pathogen showed a positive result of 28.6% in the conservative group and 20.8% in the surgical group, consistent with previous studies [4,14]. Blood and joint fluid aspirate samples should be obtained before the administration of antibiotics, but certain patients were transferred from the local hospital after the administration of variable doses of antibiotics. Previous studies showed the positive rate of bacterial culture taken from bone and joint samples to be 40%-50%, but in our result, this was 83.3% [15]. A probable explanation for this difference is the fact that multiple samples were obtained for culture and sensitivity during the surgery.

X-rays were routinely taken to exclude fracture and malignancy, and were usually inconclusive for the diagnosis of osteomyelitis at an early stage. Ultrasound is highly sensitive in detecting joint effusions [4,16]. Sometimes, ultrasound-guided aspiration was performed at our institute to obtain the samples. MRI is the preferred choice of imaging modality, with high sensitivity and specificity [16]. Therefore, MRI was performed at our institute for patients with suspected acute osteomyelitis. However, the computed tomography (CT) scan was usually avoided in the neonates, as CT is inconclusive in cartilaginous bone and exhibits high radiation exposure.

In the classical paradigm, osteomyelitis usually occurs at the metaphysis. However, due to the unique anatomy in the neonatal long bones, osteomyelitis usually occurs at the epiphysis, resulting in a high probability of septic arthritis [17,18,19,20]. This warrants prompt treatment to avoid catastrophic outcomes, including epiphyseal cartilage destruction, growth disturbances, and functional impairment [16]. Therefore, early surgical intervention was proposed at our institute.

No single test could confirm or rule out septic arthritis or osteomyelitis [5]. A combination of careful history, physical examination, imaging, laboratory tests, and aspiration/biopsy is typically required to make a definitive diagnosis [5]. Ultrasound is useful for assessing effusion in suspected septic arthritis. If increased fluid is identified, arthrocentesis can be performed under image guidance [21]. As for osteomyelitis, PET/CT has been described as superior to MRI in monitoring response to treatment, since it is better at distinguishing between ongoing infection and reparative activity [22]. However, PET/CT is not accessible in average hospital. Higher sensitivity for both diagnoses was found when combining ESR and CRP tests [13].

Antibiotic therapy is the cornerstone of the treatment of neonatal osteomyelitis. Before identification of the causative organism, empiric therapy was implemented. Empiric antimicrobial treatment should cover Staphylococcus aureus, group B Streptococci, and gram-negative bacteria [23]. Therefore, oxacillin or linezolid, combined with cephalosporin, was used at our institute. After the isolation of the causative pathogen, antibiotic therapy should be adjusted based on the culture and sensitivity profile [23]. In our study, the most common microbes were Staphylococcus aureus; only one patient in early surgery group demonstrated Gram-negative bacteria. Therefore, the empiric antimicrobial of Gram-positive bacteria, in accordance with our local epidemiology, was the primary treatment in our institution. Empirical antibiotics in accordance with local epidemiology should be implemented in other institutions before the identification of causative pathogen.

The transition from intravenous to oral antimicrobial therapy remains debatable. There is a growing literature advocating a shorter course of IV antibiotics, followed by oral antibiotics [24,25,26,27,28]. In this study, inflammatory markers changed faster in the surgical group, and a shorter course of IV antibiotic therapy was implemented in the surgical group. For oral antibiotic therapy, a shorter course was also observed in the surgical group. However, these findings should be interpreted with caution, because we believed that surgery leads to a better eradication of pathogens in this series without corroborative evidence.

The management of neonatal osteomyelitis requires a multidisciplinary approach. Pediatricians, radiologists and orthopedic surgeons must act promptly for early diagnosis. In this study, the early surgical intervention resulted in a higher rate of bacterial identification, leading to a shorter hospital stay and a lower rate of growth disturbances. Isolation of the causative microorganism is conducive to accurate antimicrobial therapy; a shorter length of hospital stay reduces the risk of nosocomial infection and lowers the emotional and financial burden on the family members; a low risk of growth disturbance is possibly due to the early drainage of pus and decompression of intra-articular pressure during surgery. Concomitant septic arthritis was not uncommon in epiphyseal osteomyelitis in neonates; therefore, early surgical intervention was warranted in this special group of patients.

Hip joint infection usually results in an unsatisfactory outcome and, therefore, requires aggressive treatment. In contrast, the outcome of shoulder joint infection is better [29,30]. Still, the easy extension into epiphyseal cartilage and concomitant septic arthritis in the shoulder joint warrants early surgical treatment for neonatal humeral osteomyelitis [31,32].

There were several limitations to our study. Firstly, it was a retrospective study with a modest sample size, and the results should be interpreted with caution; secondly, long-term follow-up on the growth and development has yet to be investigated. Therefore, we are not able to offer management advice on the long-term complications, such as a physis compromise. The assessor was the surgeon in charge, and this might cause bias. Furthermore, there is no validated function evaluation form for shoulder joint in neonate and infant, and only ROM was recorded in our study [33]. The subdivision of surgical group was arbitrary, and 3 days happened to be the length from Friday to Monday, because only emergency surgeries were allowed during the weekend in our hospital. To our knowledge, this is the first study that solely discusses humeral epiphyseal osteomyelitis in neonates. In the future, a prospective multiple center collaboration might produce a more convincing conclusion.

## 5. Conclusions

Surgical treatment for neonatal humeral epiphyseal osteomyelitis demonstrated significantly higher rates of positive culture for the pathogen, a shorter course of intravenous oral antibiotics, and lower incidence of growth abnormality than conservative treatment. In our institution, most of the culture outcome was Gram-positive bacteria, and early surgical treatment was recommended, as it led to a better outcome than delayed surgical groups. Empirical antibiotics should be tailored according to the epidemiological characteristics of local virulent bacteria before the identification of the causative pathogen.

## Figures and Tables

**Figure 1 children-09-00527-f001:**
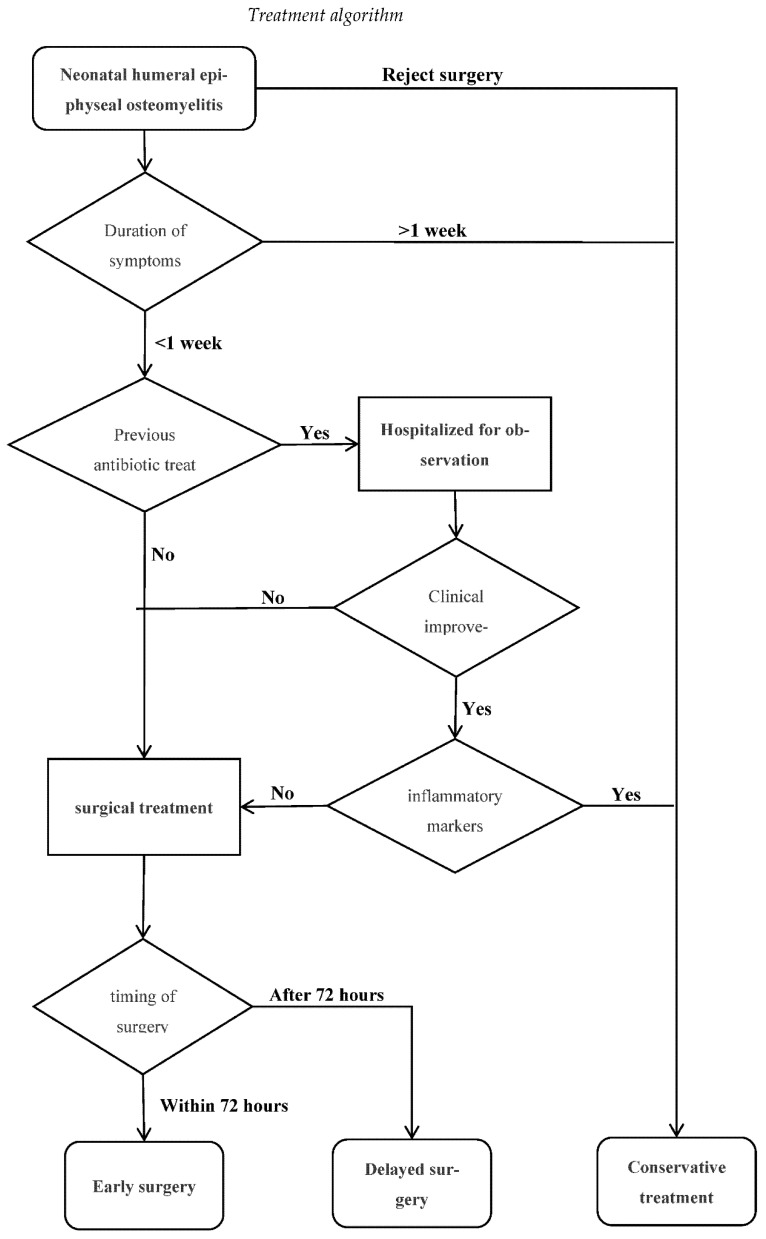
Diagnostic algorithm.

**Figure 2 children-09-00527-f002:**
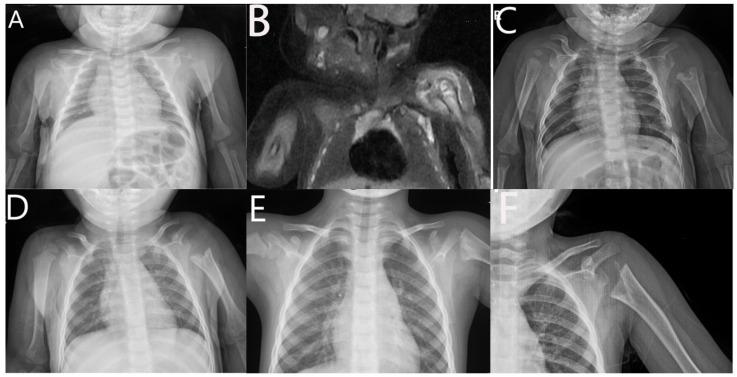
Left humeral epiphyseal osteomyelitis and septic arthritis of 25-d girl. (**A**). AP view X-ray of shoulder joint. (**B**). MR image of shoulder joint. (**C**). AP view X-ray of shoulder joint at 3rd month follow-up. (**D**). AP view X-ray of shoulder joint at 13th month follow-up. (**E**). AP view X-ray of shoulder joint at 3rd year follow-up. (**F**). AP view X-ray of shoulder joint at 4th year follow-up.

**Figure 3 children-09-00527-f003:**
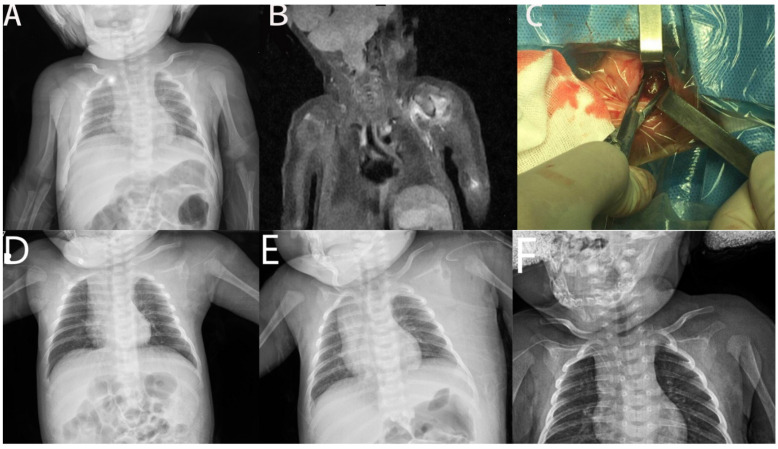
Humeral epiphyseal osteomyelitis and septic arthritis of shoulder joint of 12-day girl. (**A**). AP view of shoulder joint. (**B**). MR image of shoulder joint. (**C**). Intraoperative image of decompression and drainage. (**D**). AP view of shoulder joint at 1st month follow-up. (**E**). AP view of shoulder joint at 3rd month follow-up. (**F**). AP view of shoulder joint at 12th month follow-up.

**Figure 4 children-09-00527-f004:**
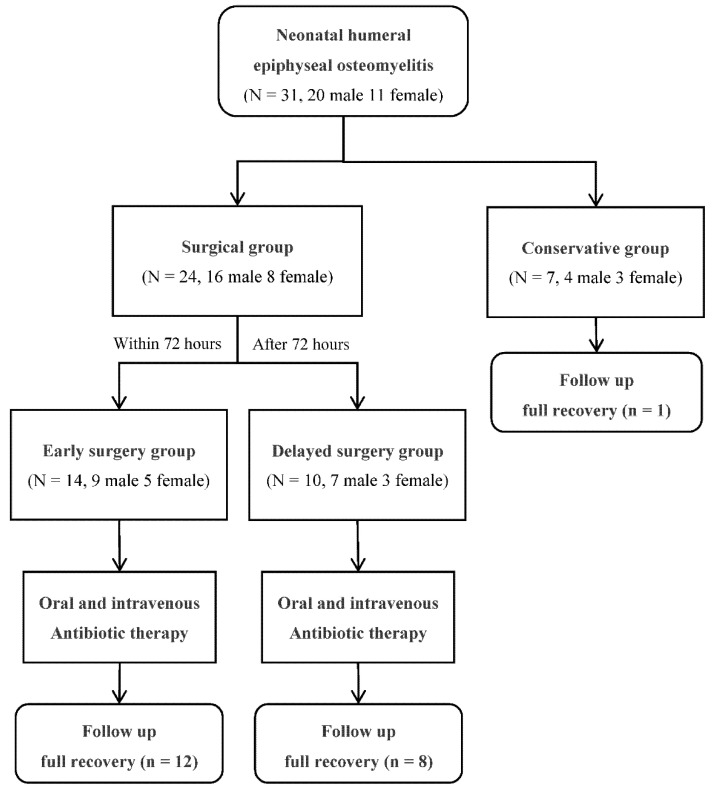
Flowchart of this study.

**Table 1 children-09-00527-t001:** Characteristics of the patients.

Parameters		Non-Surgical (*n* = 7)	Surgical (*n* = 24)	*p* Value
Age (days)		19.6 ± 6.3	18.2 ± 5.5	0.641
Sex	male	4	16	0.654
	Female	3	8	
From onset to admission (d)		11.0 ± 1.9	2.4 ± 1.6	<0.001 *
Leukocyte (10^9^/L)		17 ± 4.9	20.6 ± 6.4	0.157
Neutrophil (10^9^/L)		15.0 ± 6.7	15.5 ± 5.1	0.626
Lymphocyte (10^9^/L)		5.1 ± 1.0	5.6 ± 1.3	0.334
Platelet (10^9^/L)		411.9 ± 62.6	387.1 ± 60.7	0.407
CRP (mg/L)		44.7 ± 22.4	56.5 ± 19.9	0.272
ESR (mm/h)		31.9 ± 7.4	38.3 ± 13.0	0.133
Predisposing factors		1 (14.3%)	4 (16.7%)	0.906

d = day; * < 0.05.

**Table 2 children-09-00527-t002:** Clinical outcome of the patients.

Clinical Outcomes			Non-Surgical (*n* = 7)	Surgical (*n* = 24)	*p* Value
Culture	Blood		2 (28.6%)	5 (20.8%)	0.682
	Pus/aspirate		1 (14.3%)	17 (70.8%)	<0.001 *
Length of hospital stay (d)			45.6 ± 2.4	24.8 ± 2.8	<0.001 *
IV antibiotics (d)			45.6 ± 2.4	24.8 ± 2.8	<0.001 *
Oral antibiotics (d)			27.7 ± 1.2	19.1 ± 5.3	<0.001 *
Full recovery			1 (14.3%)	20 (83.3%)	<0.001 *
Length Discrepancy, cm			0.7 ± 0.5	0.4 ± 0.2	<0.001 *
ROM of shoulder		Flexion	130 ± 14.4	160 ± 16.2	<0.001 *
		Extension	30 ± 12.1	45 ± 12.4	<0.001 *
		Abduction	95 ± 11.4	146 ± 14.4	<0.001 *
		Adduction	40 ± 10.4	60 ± 12.4	<0.001 *
		IR	40 ± 11.7	55 ± 12.5	<0.001 *
		ER	60 ± 12.4	75 ± 11.4	<0.001 *
Normal Radiograph			1 (14.3%)	20 (83.3%)	<0.001 *

ROM = range of motion; IR = internal rotation; ER = external rotation; d = day. * < 0.05.

**Table 3 children-09-00527-t003:** Parameters of surgical intervention.

Parameters		Early Surgery (*n* = 14)	Delayed Surgery (*n* = 10)	*p* Value
Age(days)		19.9 ± 4.7	15.8 ± 5.6	0.089
Sex	male	9	7	0.792
	Female	5	3	
From onset to admission (d)		1.3 ± 1.1	3.9 ± 0.7	<0.001 *
From admission to surgery (d)		0.43 ± 0.49	0.5 ± 0.5	0.744
From onset to surgery (d)		1.7 ± 1.2	4.4 ± 0.8	<0.001 *
Leukocyte (10^9^/L)		20.5 ± 6.9	20.9 ± 5.6	0.887
Neutrophil (10^9^/L)		15.5 ± 4.8	15.5 ± 5.4	0.993
Lymphocyte (10^9^/L)		5.4 ± 1.3	5.8 ± 1.2	0.551
Platelet (10^9^/L)		371.4 ± 64.8	409.0 ± 46.4	0.127
CRP (mg/L)		55.9 ± 19.4	57.3 ± 20.6	0.877
ESR (mm/h)		37.9 ± 12.5	38.8 ± 13.7	0.881
Predisposing factors		2 (14.3%)	2 (20.0%)	0.739

d = day; * < 0.05.

**Table 4 children-09-00527-t004:** Clinical outcome of surgical intervention.

Clinical Outcomes			Early Surgery (*n* = 14)	Delayed Surgery (*n* = 10)	*p* Value
Culture	Blood		3 (21.4%)	2 (20.0%)	0.919
	Pus/aspirate		11 (78.6%) ^#^	6 (60.0%)	0.316
Length of hospital stay (d)			23.4 ± 2.4	26.9 ± 1.9	<0.01 *
IV antibiotics (d)			23.4 ± 2.4	26.9 ± 1.9	<0.01 *
Oral antibiotics (d)			15.6 ± 3.5	24.1 ± 2.7	<0.01 *
Full recovery			12 (85.7%)	8 (80.0%)	0.739
Length Discrepancy, cm			0.4 ± 0.17	0.35 ± 0.21	0.354
ROM of shoulder		Flexion	160 ± 15.9	159 ± 16.2	0.687
		Extension	45 ± 12.7	44.5 ± 12.6	0.778
		Abduction	146 ± 13.7	145 ± 14.2	0.232
		Adduction	60 ± 12.1	58 ± 11.4	0.114
		IR	55 ± 11.9	54 ± 12.5	0.089
		ER	75 ± 10.8	75 ± 11.7	0.078
Normal Radiograph			12 (85.7%)	8 (80.0%)	0.739

d = day; * < 0.05. ^#^ Only one patient in early surgery group demonstrated Gram-negative bacteria of culture outcome.

## Data Availability

Data sharing is not applicable to this article as no datasets were generated or analyzed during the current study.

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
