# Peer review of "Is Early Surgical Intervention Necessary for Acute Neonatal Humeral Epiphyseal Osteomyelitis: A Retrospective Study of 31 Patients"

_children, 2022, doi:10.3390/children9040527_

Round 1
Reviewer 1 Report
Congratulation for the good paper. I agree totally with your position and your conclusions
Author Response
Thanks for your valuable advice. The manuscript has been reviewed and revised by an English native medical practitioner.
Reviewer 2 Report
This is a well written manuscript on a topic for which additional information is needed. There are several things the authors can do to improve the value of this manuscript and expand its generalizability/utility for others.
- Provide more detail on how you defined the presence of epiphyseal osteomyelitis.
- Provide detail on the presence or absence of associated metaphyseal osteomyelitis with and without evidence of physeal injury.
- How many of your cases arose from primary bacterial arthritis versus from initial osteomyelitis?
- What were your definitions/criteria for lack of improvement used to justify surgical intervention? (Answering this would simply help readers understand--I am not concerned that you did not use appropriate criteria.)
- Did you have a protocol for guiding clinical decision-making? If so, would be helpful to provide as a supplementary figure.
- How many of the 31 children had a positive culture (joint fluid, blood, bone, etc) or molecular test for a microbial etiology?
- What were the most common microbes detected?
- Given your small sample size, it is of course difficult to detect a lot of potentially meaningful differences between your subgroups. The primary value of your results truly relates to the focus on epiphyseal involvement of a specific bone (humerus) in a specific age group (neonates). Can you clarify that the focus is on proximal humerus (shoulder), or do your cases also include distal humerus (elbow)? My presumption is proximal/shoulder, but just make this clear either way.
Author Response
Thanks for you valuable advises. Please see the attachment.
